# The Influence of Dietary Supplementation with Dried Olive Pulp on Gut Microbiota, Production Performance, Egg Quality Traits, and Health of Laying Hens

**DOI:** 10.3390/microorganisms12091916

**Published:** 2024-09-20

**Authors:** Anna Dedousi, Charalampos Kotzamanidis, Andigoni Malousi, Virginia Giantzi, Evangelia Sossidou

**Affiliations:** 1Veterinary Research Institute, Hellenic Agricultural Organization, DIMITRA, 57001 Thessaloniki, Greece; kotzam@elgo.gr (C.K.); vgiantzi@elgo.gr (V.G.); sossidou@elgo.gr (E.S.); 2Laboratory of Biological Chemistry, Medical School, Aristotle University, 54124 Thessaloniki, Greece; andigoni@auth.gr

**Keywords:** gut microbiome, layer chickens, olive paste, egg characteristics, biochemical parameters, productivity

## Abstract

This study examines the dietary effect of dried olive pulp (OP) on the overall performance, egg quality, health, and gut microbiota of laying hens during a 36-week trial. A total of 180 Isa Brown layers, aged 23 weeks, were assigned to 15-floor pens and divided into three feeding groups (CON, OP4, and OP6) based on the dietary level of OP. Egg quality and biochemical parameters were assessed in 39- and 59-week-old hens. Fecal samples were collected for microbiota analysis. Data were analyzed with an Analysis of Variance. The percentage of broken eggshells was found to be 15–34% lower in the OP groups compared to the CON groups. At 59 weeks of age, a significant reduction in shell thickness was observed in the CON eggs compared to the OP eggs (*p* < 0.05). At 39 weeks of age, OP6 eggs had the darkest yolk color of all groups (*p* < 0.05). Fecal microbial diversity was affected only by hens’ age. However, an enrichment in bacterial species belonging to the genera *Megasphaera* and *Megamonas* was found in the OP groups at 59 weeks of age. Our results demonstrate that OP feeding beneficially affects egg quality and promotes the proliferation of bacteria involved in the degradation of complex plant compounds, potentially contributing to the overall health of the gut microbiota.

## 1. Introduction

Modern poultry farming is a well-organized and globally expanding sector, driven by factors such as population growth, increased purchasing power, and urbanization. Eggs, which are considered to be a cost-effective source of animal protein, are recognized as a complete and highly nutritious food that offers significant health benefits to consumers [1]. The average per capita consumption of eggs is forecasted to reach 6.2 kg in 2024, with the egg market expected to grow by 5.2% in 2025 [2]. The rising global demand for eggs necessitates enhanced laying persistence and egg quality stability. Consequently, the layer sector has transitioned towards extended laying cycles, expecting hens to lay up to 500 eggs in a 100-week cycle [3,4]. However, the prolonged, high-intensity laying compromises gut health, hepatic metabolism, and ovarian immune function due to the metabolic pressure on the birds to meet their needs, resulting in poor egg quality late in production and significant financial losses [5,6]. Efforts to improve egg quality and hen productivity through genetic tools and biotechnology face challenges due to their time-consuming nature and high associated costs [7]. Thus, more practical and cost-effective solutions, such as appropriate dietary formulas, should be employed to address these issues.

At the same time, extensive research highlights the central role of the gut microbiota in maintaining intestinal health [8]. Previous studies in laying hens have demonstrated the beneficial impact of gut microbiota on production performance [9,10], egg quality [11,12], nutrient utilization [13,14], and intestinal barrier [8,15]. The aforementioned studies have indicated important interactions between gut microbiota and nutritional responses and physiological health in laying hens. Beyond directly impacting egg quality and safety through the intestinal–oviduct–egg transmission route, intestinal microbiota and its metabolites, such as short-chain fatty acids (SCFAs), bile acids (BAs), and tryptophan derivatives, play indirect roles in regulating egg quality through the microbiota–gut–liver/brain–reproductive tract axis [16]. For instance, SCFAs have been demonstrated to impact calcium utilization and deposition, regulating systemic calcium metabolism homeostasis and thereby enhancing eggshell quality [12]. Additionally, bile acids (BAs) generated by the intestinal microbiota influence lipid metabolism and intestinal absorption, contributing to the improvement in egg yolk color [14,17]. Recent studies have also indicated that dietary interventions involving probiotics or tea polyphenols can enhance albumen quality by reshaping the composition of intestinal microbiota and its metabolites, thereby regulating the antioxidant status and metabolic functions of the liver [5,11,17]. Consequently, the gut microbiota emerges as a potential target for future nutritional strategies aimed at modulating egg quality and safety. Moreover, it is acknowledged that host genes and environmental factors, with diet being a primary factor, determine the composition of the microbiota [18]. Therefore, dietary modifications have the potential to alter the gut microbiota.

One of the nutritional strategies implemented to ameliorate egg quality and hens’ productivity as well as maintaining gut health in poultry is the incorporation of plant-derived feed supplements rich in bioactive compounds with anti-microbial, antioxidant, and anti-inflammatory properties [12,19,20]. The ban on the use of antibiotics as growth promoters in animal nutrition has further fueled this approach [21] by boosting global efforts to find alternatives to antibiotics [22]. Olive pulp (OP), a by-product of the olive oil extraction process, stands out as one such plant-derived feed additive. In the last few years, the increased interest in utilizing agricultural byproducts as animal feeds aligns with the modern concept of the circular economy [23]. Enriched with essential fatty acids, polyphenols, and beneficial compounds like oleuropein and tyrosol, OP has been incorporated into poultry diets to improve poultry health and welfare, birds’ performance, and product quality [23,24,25,26,27,28].

Besides its beneficial effects on poultry performance, health, and product quality, researchers are increasingly focusing on olive pulp’s impact on the gut microbiome. Currently, the majority of these investigations have focused on broiler chickens [20,26,29,30,31] and very few of them in laying hens [19]. However, layers have a longer lifespan and differ significantly from broilers in terms of genotype, nutritional requirements, and rearing environment under commercial conditions [32]. As a result, the gut microbiota composition differs between these two poultry lines [33].

In our prior research, we assessed the prolonged dietary influence of dried olive pulp (OP) on the production performance, egg quality, health, and welfare of laying hens raised in a cage system, obtaining noteworthy and encouraging outcomes [25]. In a separate investigation, it was determined that the inclusion of OP in the diet of broilers does not have adverse effects on gut health [26]. However, with the laying hen housing system progressively transitioning to non-cage systems due to legislative framework and public concerns about animal welfare, a significant impact on hen productivity, egg quality and safety, gut microbiota, and producers’ income is anticipated. Considering these factors and the existing knowledge gap on the dietary impact of OP on laying hens’ gut microbiota, we conducted the current study in cage-free laying hens. The objective of this study was to investigate the influence of dietary supplementation with dried olive pulp on the gut microbiota of laying hens by using the most advanced and up-to-date metagenomic analysis methods and to elucidate its subsequent effects on important production traits, egg quality, and bird’s health. By exploring the complex interactions between diet, gut microbiota, and overall hen performance, we aim to provide valuable insights that could potentially reshape feeding practices in the poultry industry.

## 2. Materials and Methods

### 2.1. Ethics Statement

The experimental protocol of this study and its implemented animal care procedures were approved by the Committee for Research Ethics of Hellenic Agricultural Organization-DIMITRA (61366/11.11.2022).

### 2.2. Hens, Diets, and Experimental Design

The research took place in a Greek commercial poultry farm. One hundred and eighty (180) Isa Brown laying hens, 20 weeks of age, were randomly assigned to fifteen consecutive floor pens (12 birds/pen), after being weighed (initial body weight (BW): 1.42 ± 0.01 Kg) and tagged with special rings. Each pen was bedded with wood shaving and equipped with a 5-place nest box, a 10 kg bell feeder, 5 nipple drinkers, and a perch. The hens were vaccinated and offered a basal layer diet in mash form, which was in line with breed recommendations. Fresh water and feed were provided ad libitum throughout the study. The stocking density in each floor pen (4.08 hens per m^2^), and the equipment provided were in accordance with the provisions of EU Directive 1999/74/EC on alternative housing systems for laying hens [34]. Environmental conditions such as temperature, relative humidity, lighting, and ventilation were automatically controlled and followed the guidelines of breed management [35].

After a twenty-one-day adaptation period, three feeding groups were formed based on the incorporation rate of olive pulp in the birds’ diet, namely CON, OP4, and OP6. The CON group, which served as a control, was fed the basal layer diet provided to the birds during the adaptation period. The OP groups were fed with the CON diet supplemented with 4% and 6% olive pulp (OP4 and OP6, respectively). Each group consisted of 60 hens, with 5 replicate-floor pens/group and 12 hens/replicate-floor pen. The dried OP used in this study was a commercial product in flour form, which replaced a quantity of soymeal and of soya oil in the CON diet (Table 1). The OP was produced during the olive oil extraction process, in which the fruit (pulp and kernels) was mechanically pressed. The viscous suspension was subjected to a two-phase separation process and olive oil was separated from olive pulp by centrifugal methods. The olive paste is a semi-solid residue with the characteristics of a thick pulp with a moisture content (45–75%). For the OP production, the supplied company has a production line consisting of a tank for the collection of the raw material, a conveyor belt for the transfer of the mixture to the drier and a drier that works naturally at low temperatures. An appropriate system was used to separate the OP from the stones. The final grinding of the OP into flour was carried out in a roller mill.

The formulated rations were isonitrogenous and isocaloric (Table 2). Table 3 and Table 4 present the nutritional analysis and fatty acid profile of OP used in this study. The duration of the trial was 36 weeks. The nutritional analysis of the items in Table 2 and Table 3 was performed using the methods described in the Nielsen Food Analysis Manual [36].

### 2.3. Bird Productivity and Egg Quality

All birds were individually weighed at 20, 39, and 59 weeks of age. From 26 to 59 weeks of age, the feed intake was measured on a weekly basis in each replicate by weighing the amount of distributed feed and the amount of leftover and scattered feed and then calculated as the average daily feed intake per hen (ADFI). The number of eggs produced and those with soiled shells or defects (broken, cracked, or no shell) was recorded daily per replicate pen. Individual egg weights were recorded on a weekly basis for each replicate. Mortality records were kept daily. Hen day–egg production % (HDEP = total number of eggs produced on a day/number of hens present on that day) × 100, average daily feed intake (ADFI = cumulative feed intake/(number of birds x number of days)), egg mass (egg mass = (HDEP × egg weight)/100), and feed conversion ratio (FCR = g feed/g egg mass) were calculated for each replicate of each dietary treatment. 

A total of 45 eggs (15 eggs/group, 3 eggs/replicate) were randomly selected at 39 and 59 weeks of hens’ age (90 eggs in total) in order to determine some internal (albumen and yolk weights and ratio %, Haugh unit, yolk color, albumen and yolk height and pH, and yolk diameter and index) and external (shape index; eggshell weight, ratio %, and thickness; shell breaking strength; and deformation) egg quality traits. Initially, egg shape index ((width/height) × 100) was estimated from the height and width of each egg using an electronic caliper. An Egg Quality Testing Kit (Complete egg testing kit, Stable Micro Systems, Surrey, UK) was used to assess the whole egg weight, shell breaking strength and deformation, shell thickness, albumen height, and Haugh unit values. The yolk, white (albumen), and shell of the broken egg were weighed to the nearest 0.01g on a digital balance. Subsequently, the percentages of albumen ((albumen weight/egg weight) × 100), yolk ((yolk weight/egg weight) × 100), and shell ((shell weight/egg weight) × 100) in each egg were assessed. The height, width, and length of the yolk were measured using a tripod micrometer and an electronic caliper, and the yolk index ((height/diameter) × 100) was then determined. The Roche yolk color fan was used to assess the color of the yolk on a scale from light yellow (1) to deep orange (15). A waterproof pH meter was used to measure albumen and yolk pH. 

### 2.4. Blood Biochemical Parameters 

Blood samples for the estimation of certain biochemical parameters were taken at 39 and 59 weeks of age from 3 randomly selected hens per replicate (90 blood samples in total, 15 per group per sampling period). Around 2 mL of blood was collected from each hen’s brachial vein into plastic vacuum tubes (BD Vacutainer^®^ SSTTM II Advance, Becton Dickinson, NJ, USA). After clotting, the serum was centrifuged (3000× *g*, 15 min), transferred to plastic vials, and sent on ice to an ISO-certified veterinary laboratory for analysis. The following biochemical parameters were evaluated in the blood serum of the birds: triglycerides, cholesterol, gamma-glutamyl transferase (G-GT), aspartate aminotransferase (AST), blood urea nitrogen (BUN), uric acid, and glutamate dehydrogenase (GLDH). The analysis was performed using an automatic biochemical analyzer (Advia^®^ 1800 chemistry analyzer—Siemens Healthineers Headquarters, Erlangen, Germany). Commercially available diagnostic kits were used.

### 2.5. Microbiota Analysis

#### 2.5.1. Sample Collection and DNA Extraction

When the birds were 22 weeks old and before adding the olive paste to their diet, 3 samples of fresh feces (approximately 1 g each) were randomly collected from 3 floor pens (1 sample per floor pen) for microbiota analysis. At 39 and 59 weeks of age, 9 fresh fecal samples were collected at each time point from three pens (randomly selected) from each group for microbiota analysis (3 samples per group, 1 sample per pen floor). At each time point, approximately 1 g of fresh feces/pen—on the floor, collected with plastic gloves at 5 different locations in each pen (4 corners and the center)—was placed in sterile plastic tubes. After collection, the samples were frozen at −20 °C prior to DNA extraction. Total genomic DNA was isolated from approximately 250 mg of fecal contents using the Quick-DNA fecal/soil Microbe Microprep Isolation Kit (Irvine, CA, USA) according to the manufacturer’s protocol.

#### 2.5.2. Sequencing Data Analysis

Metagenomic samples were split into three groups depending on the time and treatment. The CON group included three untreated/control samples, one at each time point; the OP4 group had nine samples treated with low-dosage olive paste (OP4) and split equally into the three time points; and the OP6 group included nine samples treated with increased olive paste dosage (OP6), split also equally into the three time points of experimentation. Shotgun metagenomic sequencing ran on Illumina NovaSeq 6000 S4 using 150 bp paired-end reads. fastp [37] was used to trim adapters and to retain reads that met the Q30 mean quality for 100% of the called bases. Kraken v.2.1.2 was used to implement k-mer-based taxonomic classification [38], and KrakenTools (https://github.com/jenniferlu717/KrakenTools, accessed on 15 July 2024) was used to combine kraken-style reports into a single report file for each taxonomy class [39]. Metagenomic samples were comparatively analyzed on the phylum, genus, family, and species levels after transforming the original count-based taxonomic assay data into relative abundances. Vegan 2.6.4 [40] was used to measure the alpha diversity indices for the heterogeneity and distribution of species abundances in each sample.

Principal Coordinate Analysis (PCoA) was applied as an unsupervised ordination method to analyze the beta diversity using the Bray–Curtis index on the relative abundances. Distance-based redundancy analysis (dbRDA) was used as a supervised method to complement PCoA using Euclidean distances. PERMANOVA was used to assess the importance of each variable on the similarity between samples. Species with monotonic longitudinal increases/decreases in their relative abundance were detected using ANCOM-BC2 v.2.4.0, which enables multi-group comparisons and repeated measurements [41]. In the longitudinal setup, a species is considered monotonically enriched/depleted if one of the following criteria is true: (a) the log fold change of OP4 (lfc_GroupOP4) is <−1, and the log fold change of OP6 (lfc_GroupOP6) is less than the log fold change of OP4-1 (lfc_GroupOP4 <−1 and lfc_GroupOP6 < lfc_GroupOP4-1); (b) the log fold change of OP4 is >1, and the log fold change of OP6 is greater than the log fold change of OP4 +1 (lfc_GroupOP4 > 1 and lfc_GroupOP6 > lfc_GroupOP4 + 1) (the longitudinal setup was used in the results). Data wrangling, community index summarization, and taxonomic data analyses were implemented using package mia 1.10.0 [42]. The analysis was implemented in R 4.3.2 using Rstudio IDE v.2023.09.1, and the results were visualized using the functions of the ggplot2 library v.3.4.4 [43]

### 2.6. Statistical Analysis 

The statistical software Jeffreys’ Amazing Statistics Program JASP (v 0.16.4; JASP Team, 2022) was used for data analysis [44]. The normality of the data was tested using the Shapiro–Wilk test, and homogeneity of variance was assessed using Levene’s test. For the comparison of the average values for birds’ final BW, HDEP %, EW, FI, egg mass, FCR, and percentage of eggs with broken and dirty shells, one-way ANOVA was used. For the comparison of the average values of internal and external egg quality traits and biochemical parameters evaluated among dietary treatments at 39 and 59 weeks of age, group and hen age were used as fixed factors. A post hoc analysis was carried out using Tukey’s test. Where the distribution was not normal, comparisons were made using Kruskal–Wallis and Mann–Whitney U non-parametric tests at a significance level of *p* ≤ 0.05.

## 3. Results

### 3.1. Bird Productivity

The percentage of eggs with broken shells was significantly affected (*p* < 0.05) by the addition of OP in the diet of laying hens, as shown in Table 5. On the other hand, similar egg weight, egg mass, final BW, HDEP %, ADFI, FCR, and percentage of eggs with dirty shells were found among the dietary groups (*p* > 0.05). 

Compared to the CON hens, the OP hens produced eggs with a lower percentage of broken shells. However, significant differences were only observed between the CON and OP4 groups. All groups had the same mortality rate (1.67%).

### 3.2. Egg Quality Traits

The data in Table 6 display the results for some internal and external egg quality traits that were assessed in samples of eggs collected from hens in all groups at 39 and 59 weeks of age. Group, age, and group × age effects are shown. Data analysis revealed a significant effect of age on egg weight (*p* < 0.05). In particular, older hens (59 weeks of age) produced significantly (*p* < 0.05) heavier eggs than the younger hens (39 weeks of age). However, no group (*p* > 0.05) or group × age effects (*p* > 0.05) were observed on egg weight. Egg width was significantly affected by group and by the interaction of group × age (*p* < 0.05); OP4 eggs had a significantly (*p* < 0.05) lower egg width than the CON and OP6 eggs. Furthermore, in the OP4 group, younger hens produced eggs with greater width compared to older hens (*p* < 0.05). In addition, OP4 eggs had a significantly lower width (*p* < 0.05) than CON and OP6 eggs at 59 weeks of age. On the other hand, egg length was not affected by group, age, or their interaction (*p* > 0.05). As shown in Table 6, shape index was significantly affected by the interaction of group × age (*p* < 0.05). More specifically, in the OP4 group, younger hens produced eggs with a significantly higher shape index compared to older hens (*p* < 0.05). In addition, at 59 weeks of age, CON hens produced eggs with a significantly higher (*p* < 0.05) shape index than the OP4 group.

A significant (*p* < 0.05) age effect was observed on SBS and shell deformation; both parameters were significantly (*p* < 0.05) reduced with increasing age of the laying hens (Table 6). From the individual components of the eggs, albumen weight and shell ratio (%) were not affected by group, age, or their interaction (*p* > 0.05). The albumen ratio (%) was found to be significantly increased (*p* < 0.05) in eggs produced by younger birds (39 weeks of age) compared to that of eggs produced by older hens (59 weeks of age). The CON hens produced eggs with significantly (*p* < 0.05) higher yolk weight and yolk ratio (%) compared to those found in eggs from the OP groups. The same significant (*p* < 0.05) group effect on yolk weight and ratio (%) was observed in the eggs produced by the younger CON hens, but the differences were only numerical (*p* > 0.05) in the eggs produced by the older CON hens. Furthermore, a significant group × age effect (*p* < 0.05) was observed in the CON group for yolk ratio (%), with significantly (*p* < 0.05) higher values recorded in the eggs from younger birds compared to older hens. Shell weight was significantly (*p* < 0.05) affected by hen age, with the lowest values recorded in the eggs from 39-week-old birds. 

There was no significant effect (*p* > 0.05) of group, age, or group × age on the albumen height, the Haugh unit, or the yolk height (Table 6). On the other hand, shell thickness and yolk color were significantly affected by group (*p* < 0.05), age (*p* < 0.05), and the interaction of group × age (*p* < 0.05). Regarding shell thickness, the results showed that older hens laid eggs with significantly (*p* < 0.05) thinner shells compared to eggs laid by younger birds and that OP eggs had significantly (*p* < 0.05) higher shell thickness compared to CON eggs. The same group effect for shell thickness was found to be significant (*p* < 0.05) in the eggs produced from 59-week-old birds but not in the eggs collected from 39-week-old hens. In addition, the age effect on shell thickness was significant in the CON group (*p* < 0.05) and numerical in the OP groups (*p* > 0.05). Yolk color assessment showed that the OP6 eggs had significantly (*p* < 0.05) darker yolk color compared to the CON and OP4 eggs. The same significant (*p* < 0.05) group effect was observed in the eggs produced by 39-week-old birds but not in those collected from older hens (*p* > 0.05). Furthermore, a significantly paler yolk color was found in the eggs collected from hens at 39 weeks of age (*p* < 0.05) compared to that found in the eggs collected from older birds. The same significant (*p* < 0.05) age effect was observed in the CON and OP4 groups but not in the OP6 group where the opposite finding was observed. 

Yolk width and yolk index were significantly (*p* < 0.05) affected by hen age (Table 6). For the yolk width, the highest values were recorded in the eggs produced from 59-week-old birds. The same significant (*p* < 0.05) age effect was observed in the OP4 group, but not in the other two treatments, where the differences in yolk width were numerical (p > 0.05). On the other hand, 39-week-old hens produced eggs with significantly (*p* < 0.05) higher yolk index than older birds. This age effect was significant (*p* < 0.05) for the OP6 group, but only numerical (*p* > 0.05) for the other two groups. According to our results, the yolk and albumen pH of the eggs were significantly (*p* < 0.05) reduced as the age of the laying hens increased. For yolk pH, the observed reduction was significant (*p* < 0.05) in the eggs from the OP4 group but only numerical (*p* > 0.05) in the eggs from the CON and OP6 groups. As for the albumen pH, the same significant (*p* < 0.05) age effect was observed only in the eggs from the CON and OP4 groups, but not in the eggs from the OP6 group. Albumen pH was also significantly affected by group (*p* < 0.05) and by the interaction of group × age (*p* < 0.05). In particular, the OP6 eggs had significantly (*p* < 0.05) higher albumen pH than the OP4 and CON eggs. The same significant (*p* < 0.05) group effect on albumen pH was observed in the eggs produced by 59-week-old hens but not in the eggs collected from younger birds (*p* > 0.05).

### 3.3. Blood Biochemical Parameters

Table 7 shows the results for the selected biochemical indicators evaluated in the blood samples collected from 39- and 59-week-old CON, OP4, and OP6 hens. The group, age, and group × age effects are shown. The serum cholesterol, triglycerides, and BUN levels were not significantly affected by the addition of olive pulp to the diet of laying hens (*p* > 0.05), hen age (*p* > 0.05), or the interaction of group × age (*p* > 0.05). There was, however, a trend (*p* = 0.065) towards lower levels of cholesterol in the serum of older hens compared with younger birds. The level of uric acid concentration was significantly (*p* < 0.05) affected by the age of the hens, with the reported value decreasing as the age of the hens increased. This reduction was found to be significant (*p* < 0.05) in the OP groups and only numerical (*p* > 0.05) in the CON hens. 

The analysis of the data showed that the liver enzyme concentrations were significantly affected by group (*p* < 0.05; G-GT, GLDH), hen age (*p* < 0.05; AST, GLDH), and the interaction of group × age (*p* < 0.05; G-GT). Specifically, a significant (*p* < 0.05) reduction in the AST levels was observed in the serum of hens at 59 weeks of age compared with younger birds. The same significant (*p* < 0.05) age effect was found in the CON and OP6 groups. However, the reduction observed was numerical (*p* > 0.05) in the OP4 group. According to the results, significantly (*p* < 0.05) higher levels of G-GT were found in the serum of the OP hens compared to the CON birds. The same significant group (*p* < 0.05) effect on G-GT was observed in 39-week-old hens but not in older hens. In addition, in the CON group, the G-GT levels were significantly (*p* < 0.05) higher in the 59-week-old hens than in the younger birds. As shown in Table 7, the serum concentration of GLDH decreased significantly (*p* < 0.05) with increasing hen age. Moreover, significantly (*p* < 0.05) higher levels of GLDH were recorded in OP4 compared to the CON hens. 

### 3.4. Dietary Effects of OP on Fecal Microbiome Structure

#### 3.4.1. Microbiome Diversity Analysis

OP supplementation had no effect on alpha diversity indices. No significant differences (*p* > 0.05) in species evenness (Figure 1) and species richness (expressed by the Chao, ACE, and Hill indices) were observed between all OP-fed groups over time (Figure 2). These results indicate that OP does not modulate the overall species heterogeneity of the fecal microbial community. This trend was also evident in the beta diversity analysis we performed to quantify the (dis)similarities between the metagenomic samples. Specifically, distance-based redundancy analysis (dbRDA) using Bray–Curtis dissimilarity was used to visualize the difference in bacterial community structure between the OP-fed groups and between the age groups of the birds. The dbRDA visualization showed a clear separation of fecal microbial communities based on sampling time point; distinct clustering patterns were observed between samples from different time points (Figure 3). On the other hand, using OP feeding as an explanatory variable, dbRDA showed no significant differences in microbial structure between the OP-fed groups; the samples in the control and OP groups were closely grouped, suggesting that the addition of OP did not alter the microbial community in the feces.

#### 3.4.2. Microbial Community Profile

We investigated the microbial composition of the feces at different stages of development and during treatment with the feed. Calculation of the relative abundance at different taxonomic levels revealed that at the phylum level, *Actinomycetota* (formerly *Actinobacteria*) was most abundant (14.7 to 62.5%), followed by *Bacillota* (formerly Firmicutes; 25.0 to 54.3%) and *Pseudomonadota* (formerly *Proteobacteria*; 4.6 to 22.3%) (Figure 4). At 39 and 59 days of age, a numerical increase in *Actinomycetota* and a decrease in *Bacillota* shifts were observed in the OP4 and OP6 diet groups compared to the CON diet; *Actinomycetota* form the dominant fecal phylum in the OP diet groups compared to the CON groups, where *Bacillota* are the predominant phylum. In the 39 and 59 weeks of age samples, the most abundant families within the *Actinomycetota* were *Dermabacteraceae* (7.6 to 17.0%) and *Brevibacteriaceae* (9.9 to 16.9%), while the most abundant families of *Bacillota* were *Lactobacillaceae* (1.0 to 31.9%) and *Enterococcaceae* (3.8 to 16.4%). At the genus level, at 39 and 59 weeks of age, the dominant groups were *Brevibacterium* (10.1 to 17.2%) and *Brachybacterium* (7.8 to 17.3%), with a numerical increase in shifts observed in the latter in dietary groups OP4 and OP6 compared to CON. Differential abundance analysis (DAA) was used to determine the differences in abundances of individual species between the three treatment groups and sampling dates. Pairwise comparisons between treatment groups at 59 weeks of age clearly showed that the relative abundance of *Megasphaera stantonii* and *Megamonas hypermegale* increased with the increase in OP dose (Figure 5).

## 4. Discussion

The present study revealed that the inclusion of 4% and 6% of OP in the diet of laying hens housed in a floor system during a laying period did not adversely affect their overall performance. The results confirm those of our previous work in terms of final hen BW, ADFI, and FCR [25]. In line with our current findings, other authors also found no dietary impact of OP on hens’ body weight [45,46,47,48], HDEP % [46,47,48,49,50,51,52], egg weight [46,47,50,52], egg mass [46,47,48,49,50,51,52], feed consumption [49,51,52,53], and FCR [46,49,53]. Contrary to our findings, increased FCR [45,47,50,51] and increased feed intake [45,46,47,50] have been previously documented in layer-fed diets containing 4–20% OP compared to those fed standard diets. In addition, previous research in caged laying hens has shown that supplementation of their diets with OP reduces egg production. [25,45,53], egg weight [25,48], and egg mass [25,45] or increases egg weight [45,49,53]. On the other hand, according to Habib et al. [28], feeding laying hens 1.5%, 3%, and 4.5% OP for 12 weeks increased total egg production, improved FCR, and had no effect on egg weight.

The different results obtained in the current trial compared to our previous research [25] regarding HDEP %, egg weight, and egg mass could be due to the different production systems used (cage vs. floor). It is well recognized that the housing system affects laying hens’ performance and the quality of produced eggs [54,55]. On the other hand, in line with our previous investigation [25], the percentage of eggs with dirty eggshells was similar between groups, indirectly suggesting that the addition of OP to the birds’ diet at the doses studied does not cause diarrhea. As expected, the percentage of dirty eggs recorded in all groups in this trial was higher than that in our earlier work apparently due to the different production systems. 

The current study showed that the addition of OP to the diet of laying hens reduced the percentage of broken eggshells by 15–34% compared to controls, confirming our previous findings [25]. Taking into account the analysis of egg quality characteristics, the lower percentage of broken eggshells recorded in the OP groups could be attributed to the increased shell thickness of the OP eggs compared to the CON eggs, as indicated by the group’s marginal means. In particular, during the second phase of the production cycle, there was a significant decrease in shell thickness for CON eggs compared to OP eggs. Consequently, this parameter differed significantly between the CON and OP groups during this phase. In addition, OP eggs had numerically higher SBS than CON eggs at this time. These results suggest a positive nutritional effect of OP on eggshell quality. Improved egg quality is vital for the egg market industry, as damaged and cracked eggs result in financial setbacks throughout the production and distribution process, amounting to as much as 8% to 11% of total egg production [56]. Furthermore, harmful bacteria can enter the cracked shell, posing a food safety concern [4]. 

At present, there is a lack of research evidence on the dietary effects of OP or other olive oil by-products on shell strength. However, several mechanisms could be involved in the results obtained, alone or in combination. Olive pulp is recognized as a rich source of minerals, including calcium, phosphorus, zinc, manganese, and copper [57]. Several research investigations have indicated that adding trace elements to the diets of laying hens may enhance both the breaking strength and mass percentage of eggshells, consequently ameliorating eggshell quality after peak production [58,59]. Thus, it could be supported that the minerals of the supplemented OP improved the structural characteristics and the quality of the eggshells. According to Zhang and Kim [60], the addition of 2% and 5% olive oil to the diet of laying hens resulted in an increase in eggshell breaking strength and shell thickness compared to control groups. These authors supported that olive oil, which is known for its ability to solubilize vitamin D, can increase calcium concentrations in eggshells. Vitamin D, as a fat-soluble vitamin, directly influences calcium absorption. Olive pulp has a high content of essential fatty acids (73% oleic acid, 13% palmitic acid, and 7% linoleic acid) as well as high residual oil [61]. Consequently, increased absorption of vitamin D from OP hens cannot be ruled out. 

The amelioration of egg quality in OP-fed layers observed in this study could also be linked to the bioactive compounds of OP such as polyphenols. Extensive research has shown that the addition of polyphenols in poultry diets at appropriate levels increase shell thickness and eggshell strength and decrease the number of broken eggs [62]. According to some authors, the mechanisms implicated in this improvement seem to involve an increase in plasma antioxidant enzymes activity, luteinizing hormone levels, and mineral content such as Ca and P [63]. On the other hand, improved egg quality during the late laying period of quails fed curcumin, which is a hydrophobic polyphenol, was attributed to improved lipid metabolism and selective regulation of the gut microbial community [64]. In addition, the improvement in egg quality documented in the OP groups could also be related to the fiber content of OP and its effect on the gut health of OP hens. The presence of non-starch polysaccharides (NSPs), particularly insoluble ones, in the cell wall of OP has been previously established [65,66]. These NSPs have been demonstrated to induce advantageous outcomes for gut health and nutrient absorption by boosting crop and gizzard activity, promoting the production of digestive enzymes, and improving bacterial fermentation in the hind gut [67,68]. Increased shell-breaking strength and thickness as well as reducing the cracked/broken egg rate following fiber supplementation in laying hens diet has been previously documented [69,70,71]. Nutrients that are not digested by the time they reach the hind gut can be used as a vital source of nutrients for commensal bacteria. Furthermore, SCFAs, the final products of fermentation of dietary fiber by the intestinal microbiota has been shown to decrease the pH within the intestinal environment, thereby improving the solubility, absorption, and deposition of calcium, leading to enhanced eggshell thickness [16]. 

Consistent with our results, Rebollada-Merino et al. [19] observed that the supplementation of laying hens’ diet with fermented defatted alperujo (FDA), a modified olive oil by-product, reduced the number of broken eggs in the supplemented group compared to controls. According to these authors, the high fiber, high-quality fat, and phenolic content of FDA improved the intestinal health of supplemented hens by modifying the intestinal structure of the pullets and aged hens, thus improving the absorptive capacity of the intestinal mucosa, and also by modifying the intestinal microbiota in favor of eggshell quality. Previous feeding trials of OP supplementation in laying hen diets, at various levels (4.5–20%) for shorter periods (up to 16 weeks), have produced controversial results on eggshell quality. Contrary to our findings, some researchers found no effect of OP on shell thickness [46,47,49,53], while others observed that control hens produced eggs with thicker shells compared to the OP groups [45,50,51]. In line with previous reports, the addition of OP in laying hens’ diet had no adverse effect on the eggshell weight and ratio % [25,46,53]. However, a number of investigators have documented an increase in shell weight and shell percentage [47,49]. On the other hand, a decrease in the shell weight of eggs produced by hens fed high levels of OP (17–20%) has also been observed [50,51]. 

The current study confirms previous findings that the egg weight increases with hen age [1,72,73] and that various eggshell quality factors, such as shell strength, thickness, and shell deformation, progressively deteriorate with hen age throughout the production cycle [74,75,76]. However, the decrease in shell thickness in the second phase of the laying period was observed only in the CON eggs, while the shells of the OP eggs maintained their good quality. Furthermore, our study showed that eggshell weight increased with hen’s age, which is in agreement with previous reports [76,77]. Consistent with our findings, Kraus and Zita [72] reported that eggshell percentage was not affected by hen age.

The research findings indicate that as hens age increases, the egg shape index typically declines [72,78]. However, in our study, we only observed this age-related impact on OP4 eggs, which may be linked to the reduced width of OP4 eggs with the progress of hen’s age. Consequently, during the second phase of the production cycle, OP4 eggs exhibited a reduced egg shape compared to the other groups (group × age effect). In line with previous studies, our egg quality analysis revealed that the inclusion of OP in the diet of laying hens had no effect on the egg shape index throughout the experimental period [25,46,51,52,53]. However, a reduction in shape index in eggs produced by OP-fed layers has been reported by other investigators [47,50].

According to internal egg quality results, supplementing laying hens’ diet with OP does not affect albumen weight and albumen percentage, confirming previous relevant studies [25,46,47]. It was also shown that the percentage of albumen decreased with increasing age of the hen, which is in line with earlier documents [77,78]. However, in contrast to our observations of a non-significant effect of hen age on albumen weight, many authors have demonstrated an increase in this parameter as laying hens get older [1,77,78]. A significant group effect on yolk weight and yolk ratio (%) was observed in this trial. In particular, the OP groups produced eggs with a lower yolk weight and percentage compared to the CON group, and this difference was found to be significant in the first phase of the production cycle but became numerical with hen age. The results obtained here are not in agreement with previous reports, according to which the addition of OP to the diet of laying hens at levels ranging from 2% to 20% had no effect on these parameters [25,46,47]. It is worth noting, however, that despite the reduction in yolk weight and ratio found in the current trial, these values remained within the ranges recorded in our previous study [25] and also within the ranges documented by other authors for Isa Brown [72]. The mechanism involved in the observed reduction in yolk weight and ratio is currently unknown, but the higher fiber content of the OP diets compared to the CON diet may be related. According to Röhe et al. [70], laying hens fed a high-fiber diet produced eggs with a lighter yolk weight (16.4 g) compared to the controls (18.2 g) at 42 weeks of age. In addition, Andrade et al. [79] observed that levels above 2.5% dietary fiber resulted in a gradual decrease in yolk percentage from the beginning to the peak of the laying period. However, further research is needed to explain our findings. With regard to the effect of hen age on yolk percentage, this study showed that the yolk proportion values of OP eggs remained constant throughout the period monitored, confirming previous reports [72,80]. On the other hand, the yolk percentage of CON eggs decreased significantly with hen age. This finding could be attributed to the numerical increase in egg weight and the significant decrease in yolk weight of the CON eggs with increasing hen age. It is generally supported that yolk percentage increases as laying hens get older [1,77].

In line with our previous study, the addition of 4% and 6% OP to the diet of laying hens housed in the floor system did not influence albumen height and Haugh unit [25]. Accordingly, other authors also found no dietary impact of OP in Haugh unit [46,47,52,53]. On the other hand, the decreased Haugh unit in eggs produced by hens fed diets with OP levels above 9% has been documented in other reports [47,49,50]. Albumen height and HU (Haugh unit) are key factors in assessing internal egg quality. It is widely accepted that Haugh unit values above 70 indicate superior freshness and overall egg quality. According to United States Standards (USDA) [81], eggs undergo grading and categorization as AA, A, and B. Grade AA eggs signify excellent quality, characterized by thick and firm whites and defect-free yolks. A grade AA egg typically has an HU exceeding 72. It is generally recognized that both albumen height and HU decline as the production cycle progresses, which negatively affects the overall egg quality [72,73,76,80]. However, in our study, eggs from all three dietary groups maintained HU values surpassing 85, thus meeting the grade AA classification.

Similar to former reports, no dietary impact of OP on yolk index was detected in this experiment [25,47,50,51,52]. Nevertheless, in other studies where the OP levels used were equal to or greater than 9%, the yolk index of the OP eggs increased [46,53]. This beneficial effect of OP was associated with its bioactive compounds like polyphenols and PUFA [46]. Furthermore, our findings demonstrate that yolk index decreases as hen age increases in accordance with earlier publications [72,74]. However, this age effect was significant only for the OP6 group. Contrarily, other authors found that the yolk index of produced eggs remained basically constant (0.41–0.45) between 30 and 70 weeks of hens’ age [80]. Similar to Andrade et al. [79], we also observed an increase in yolk width with the progression of hen age, but this effect was significant only in the OP4 group.

This study demonstrated a positive dietary effect of OP on yolk color at the incorporation rate of 6%, as indicated by the group marginal means. In particular, we observed that in the first phase of the production cycle, OP6 eggs had a significantly more intense yolk color than the other two groups. This is an important finding for both the poultry industry and consumers. It is generally accepted that consumers and egg processors prefer darker yolks [24], with color values ranging from 8 to 14 (based on the Roche yolk color fan), from moderate yellow to orange [82]. Eggs with this desired characteristic are therefore considered to be value-added products that increase the financial income of producers. In many countries and regions worldwide, consumers mistakenly associate a yellow or golden-orange yolk color with the concept of “nutritious eggs”. Conversely, a pale color is often associated with potential hen health issues or unfavorable production conditions [83]. However, it is well known that diet and management conditions play a key role for the yolk pigmentation [83]. It is established that the color of the yolk is mainly influenced by the intake of carotenoids in the diet [83] since laying hens are incapable of synthesizing xanthophylls, which are the primary pigmented carotenoids [82]. The dietary effect of OP on carotenoid metabolism is currently unknown, while the available studies investigating the effect of OP on egg yolk color have given rather inconsistent results. In line with our findings, Ibrahim et al. [61] observed that the addition of 5% and 10% OP to the diet of laying quails increased the yolk color from 9 to 10 (in the CON and OP groups, respectively). On the other hand, a higher incorporation rate of OP (16%) in the diet of laying hens aged 56 and 80 weeks for a shorter feeding period of 7 weeks resulted in paler yolk colors [50,51]. In contrast to our results, a non-significant effect of OP on yolk color when added to laying hens’ diet has been previously reported by a number of authors [25,46,47,49,52].

The darker yolk color of the OP6 eggs during the first phase of the laying period suggests that OP may have improved carotenoid bioavailability, either by increasing their absorption or their supply to the oviduct. Even though the metabolism of carotenoids in animals including birds is poorly understood, several mechanisms could be implicated in the results obtained. The absorption of carotenoids, which are fat-soluble pigments, may have been favored by the high levels of essential fatty acids and residual oil in the OP used in this study at its highest dose. It is also possible that the antioxidant properties of the OP phenolic compounds may have protected the carotenoids from oxidation. According to the available research, the use of polyphenols in poultry nutrition has been shown to ameliorate yolk color intensity and also increase the carotenoid content and the antioxidant capacity of egg yolk [62]. In addition, fiber levels in diets may modify the growth and composition of the microbiota [68]. Consequently, the darker yolk color observed in the OP6 eggs could be related to changes in the gut microflora of this group caused by the OP during the first stage of the production cycle. It has been previously demonstrated that the bile acids (BAs) generated by gut microbiota regulate lipid metabolism and the intestinal absorption, thus enhancing the color of the egg yolk [16]. Moreover, an association between yolk color and certain groups of gut microorganisms has been recently observed by some researchers [84]. Previous studies have shown that both the source and amount of fiber added to the diet of laying hens affect the yolk color of the eggs produced, suggesting that fiber may interfere with the absorption of pigments [85,86]. Specifically, darker egg yolk colors were observed in the groups fed the high neutral detergent fiber (NDF) diets compared to those fed the low NDF diets [86]. Thus, it could be supported that the higher fiber content of the OP6 diet may have contributed to some extent to the result obtained in the current study. Nonetheless, further research is required to justify our finding.

As observed in previous studies, yolk color was significantly influenced by hen age. In line with Banaszewska et al. [87], we also found that older hens in the CON and OP4 groups laid eggs with more intense yolk color compared to younger birds. However, in the OP6 group, a paler yolk color was recorded with increasing hen age (9.20 vs. 8.33). This is consistent with the findings of Drzazga et al. [80], who reported a decrease in yolk color from 7.72 to 6.82 (Roche yolk color fan) as hens progressed from 30 to 40 weeks. A similar age effect was also noticed by Padhi et al. [78] as well as by Benavides-Reyes et al. [74]. On the other hand, a number of other studies demonstrated no significant effect of hen age on yolk color [72,73,76,88]. The increase in yolk color documented in the CON and OP4 groups with hen age and the decrease in this trait recorded in the OP6 group at the same time resulted in similar yolk colors to the produced eggs among treatments at the second phase of the production cycle. 

To our knowledge, the data available in the current literature on the dietary effects of OP on albumen and yolk pH are very limited. In line with our previous report in a cage system, the incorporation of OP in laying hens at both studied levels of 4% and 6% did not influence yolk pH [25]. However, the obtained values were slightly lower (5.93, 6.08, and 6.01) than those observed in our former study (6.30, 6.22, and 6.17) for the CON, OP4, and OP6 groups, respectively, possibly due to the different housing systems used in the two trials. Despite the observed differences, yolk pH values remained within the ranges reported by other authors [89]. This experiment revealed that in the second phase of the production cycle, OP6 eggs had significantly higher albumen pH compared to the CON and OP4 groups. However, this group effect was not observed in younger hens. This result is partially in agreement with our former study [25], where data analysis of the eggs produced by 68-week-old hens showed an increased albumen pH in eggs from the OP4 and OP6 groups compared to the CON group. It is worth noting however that the albumen pH values from all three groups were similar to those recorded in our previous work [25].

In the current study, both albumen and yolk pH were affected by hen age, as previously reported by other researchers. According to Marzec et al. [88], albumen pH increased from 7.8 to 8.2 as hen age progressed from 26 to 46 weeks and then remained constant until 70 weeks of age. A similar trend was observed by these authors for yolk pH values, which initially increased from 5.6 to 6.1 and then stabilized. Vlčková et al. [90] also reported an increase in albumen pH from 8.37 to 8.89 as laying hens’ age increased from 26 to 51 weeks. On the other hand, some researchers have found no significant effect of hen age on the albumen or the yolk pH [87,91]. The findings of the aforementioned authors are consistent with our results for albumen pH in the OP6 eggs and for yolk pH in the CON and OP6 eggs. The decrease in egg albumen pH with hen age in the CON and OP6 eggs, combined with the constant value of this characteristic in the OP6 eggs, resulted in the differences observed between the treatments in egg albumen pH in older hens.

While it is important to have data to support the efficacy of new feed ingredients in improving performance and quality standards, further measures are required to ensure the safety of these ingredients for laying hens and to meet regulatory requirements. As serum biochemistry is considered to be a good indicator of the health status of poultry species [92,93], we decided to investigate the dietary impact of OP on selected biochemical indicators. In line with our former report [25], the liver and kidney function of laying hens housed in the floor system were not adversely affected by the inclusion of 4% and 6% OP in their diet. The increased levels of serum G-GT and GLDH enzymes observed in the OP groups compared to the CON hens indicate increased hepatic metabolic activity in the OP hens and could be related to the fat content of OP, which is rich in essential fatty acids and residual oil. It is worth noting, however, that the AST serum concentration in the hens of all groups were within the reported reference values of 125–210 IU/L [94], while the GLDH values in all groups also remained below the reference value of 8 IU/L reported for various poultry species [95]. In addition, G-GT levels similar to those found in this study have previously been reported by other researchers [94,96]. It has been previously stated that elevated levels of AST exceeding 275 units/L signify impairment in both liver and muscle function, with significant liver damage evident only when activity surpasses 800 units/L [94]. Taking all this into account, it could be argued that the documented increase in G-GT and GLDH levels in the OP groups in the current study is not clinically significant. In support of our findings, liver function was not impaired in laying hens fed diets containing OP, as indicated by the assessment of AST and ALT levels in the study by Al-Harthi [46]. However, increased AST serum activity in the OP-fed layers has been found by other authors [45,47]. 

Regarding serum cholesterol and triglyceride levels, our results are consistent with those of other investigators who have similarly observed no dietary effect of OP on serum cholesterol [53], triglycerides [51], or both [49,52] and confirm our previous findings for these two biochemical parameters [25]. However, contrasting observations have been reported by other researchers, indicating a decrease in serum cholesterol [51], triglycerides [53], or both items [45]. In our former study, we found increased serum uric acid levels in laying hens consuming diets containing 3%, 5%, and 6% OP compared to the CON group [25]. However, this dietary effect of OP was not demonstrated in the current work. The variability of the current results regarding the dietary effects of OP on the biochemical parameters evaluated with those obtained in our previous feeding trial could be related to the different housing systems used. The influence of the housing system on the laying hens’ blood parameters has been reported by other authors in relevant studies [97,98,99]. 

Existing research supports our findings of an age effect on some serum biochemical indicators. However, the reported fluctuations in biochemical concentrations throughout the laying period are not always consistent between studies. The main reason for this is the differences in production systems, breeds, and diets used, as well as the age of the hens in each trial. All of these factors have been shown to affect the blood parameters in laying hens [97,98,99,100]. Similar to our results, Tang et al. [101] also demonstrated a decrease in AST levels as hen age progressed from 36 to 52 weeks. In contrast, other reports have shown an increase in serum AST concentration with hen age [93,102]. According to Kraus et al. [99], AST levels remained constant throughout the laying period studied, from 34 to 50 weeks of age. Consistent with our findings, Usturoi et al. [102] reported a decrease in uric acid concentration from 7.85 to 6.89 mg/dl with increasing hen age from 35 to 55 weeks. Accordingly, Pavlic et al. [97] demonstrated a decline in uric acid levels as the hen age progressed from 22 to 47 weeks. In the later stages of the production cycle, however, these authors found that uric acid levels in the blood increased. 

Previous studies have shown that various factors such as the age of the animals; housing systems; management; diet; and use of antibiotics, probiotics, and phytobiotics can influence the composition of the intestinal microbiota in poultry [84]. Among other things, there is increasing evidence that feed supplements influence animal performance by modulating the gut microbiota [18,103]. To better understand the mechanisms underlying the beneficial effects of OP on poultry performance, health, and product quality, we focused on the ability of OP to modulate the fecal microbiota. Despite ongoing controversy as to whether the fecal microbiome accurately represents microbiomes throughout the gastrointestinal tract, the fecal microbiome has been found to have comparable diversity to the cecal microbiome [104,105].

In our study, in agreement with previous studies in which OP was used as a feed additive [26,29,30], it was found that treatment with OP did not significantly affect microbial diversity in the chickens’ feces; microbial richness and uniformity did not change with OP supplementation. On the other hand, an analysis of beta diversity using OP treatment and age of the chickens as explanatory variables showed that the microbial composition of feces was determined by age. This is consistent with previous studies that have shown that age plays a crucial role in shaping the microbial composition of chicken feces [106,107]. Furthermore, similar to the results of previously published meta-analysis reports, the results of our study showed that chicken feces were mainly dominated by *Actinomycetota* and *Bacillota*. *Actinomycetota* represent one of the largest phyla among bacteria, which play an important role in the animal gut by degrading organic matter, producing bioactive metabolites, and maintaining intestinal homeostasis [108,109], while *Bacillota* in the gut are known to be responsible for meeting the nutritional and energy requirements of animals by fermenting cellulose to SCFAs [110,111].

We used DAA to analyze the species with significant differences in abundance between the groups. Bacterial species that are characteristic of OP treatment and whose relative abundance increases with increasing treatment dose include taxa associated with dietary exposure to fermentable fibers and polyphenols, as elucidated by differential analysis. The enrichment of bacterial species belonging to the genera *Megasphaera* and *Megamonas* could be due to the targeted degradation of OP in food by these groups of bacteria, as they are known to specialize in the degradation of phenolic compounds and dietary fiber. The genus *Megamonas* is involved in the fermentation of dietary fiber and carbohydrates and produces short-chain fatty acids (SCFAs) such as acetate and propionate. These SCFAs serve as important energy sources for the host and play a crucial role in maintaining gut health by lowering the pH and inhibiting the growth of pathogenic bacteria [112,113]. On the other hand, the genus *Megasphaera* is a well-recognized butyrate-producing gut bacterium [112,114]. There is evidence that butyric acid has positive effects on the laying and hatching performance of hens [11,115]. Previous studies have also shown that the addition of butyric acid to the diet improves the absorption and utilization of minerals, including calcium [116,117], egg quality, eggshell strength [11], and yolk color [14]. In our study, the improvement in egg quality in the OP groups at 59 weeks of age can be attributed to the differential abundance of genera that increased the amount of butyrate in the cecum.

Overall, our study suggests that the addition of OP in the diet promotes the proliferation of bacteria involved in the degradation of complex plant compounds, potentially contributing to the overall health of the gut microbiota of laying hens.

## 5. Conclusions

The present study demonstrated a clear positive nutritional effect of OP on eggshell quality when added to the diet of laying hens housed in a floor system during a laying period at both studied levels, 4% and 6%. This positive effect was reflected in a reduction in the percentage of broken eggshells by 15–34% in the OP groups compared to the controls and in a reduction in shell thickness in the CON eggs compared to the OP eggs during the second phase of the production cycle. The increased number of bacterial species belonging to the genera *Megasphaera* and *Megamonas* identified in the OP groups, which are known to specialize in the degradation of phenolic compounds and dietary fiber, indicates that OP creates a favorable environment for these species within the hen’s gut, resulting in enhanced intestinal health and thus further contributing to the observed improvement in egg quality within the OP groups. Furthermore, this study demonstrated that OP has a favorable impact on egg yolk color when incorporated at a rate of 6% during the initial phase of the production cycle. The addition of OP to the diet of laying hens did not result in any alteration to the diversity of the microbial community present in the birds’ feces. The microbial composition of the feces was found to be age-dependent, with this also affecting some of the egg quality and serum biochemical parameters studied. Furthermore, no adverse effects on bird health or performance were observed at either of the OP levels studied (4% and 6%). Based on the aforementioned findings, the addition of OP to the diet of laying hens housed in an alternative housing system is strongly recommended at incorporation rates of 4% and 6% for the enhancement of egg quality and intestinal health of the hens.

## Figures and Tables

**Figure 1 microorganisms-12-01916-f001:**
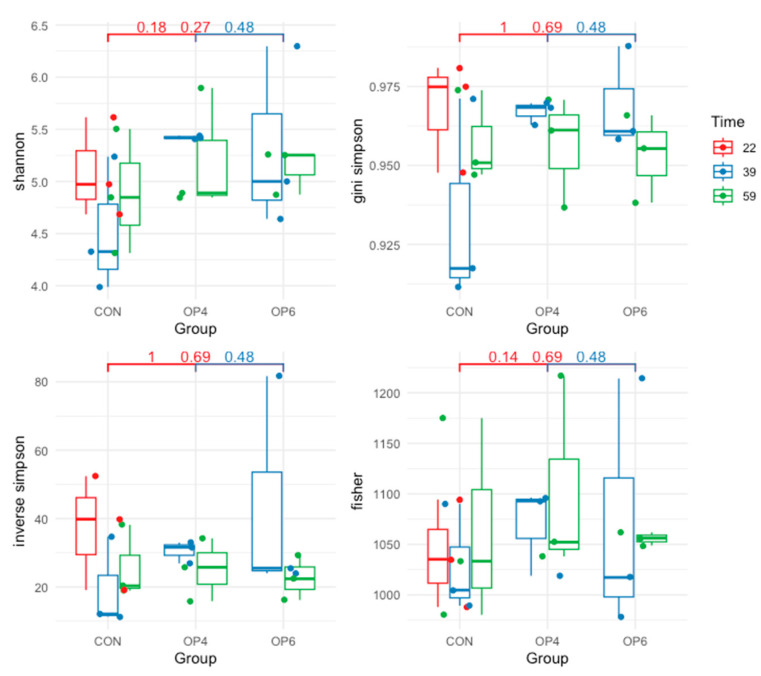
Alpha diversity indices for each treatment group (CON, OP4, and OP6) and time point (22, 39, and 59 weeks of age).

**Figure 2 microorganisms-12-01916-f002:**
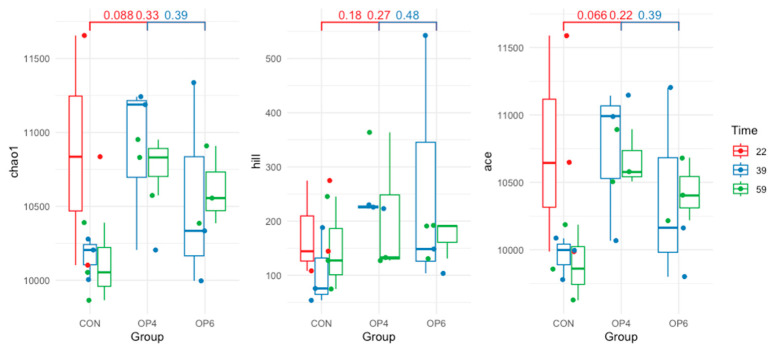
Richness indices in the species community for each treatment group (CON, OP4, and OP6) and time point (22, 39, and 59 weeks of age).

**Figure 3 microorganisms-12-01916-f003:**
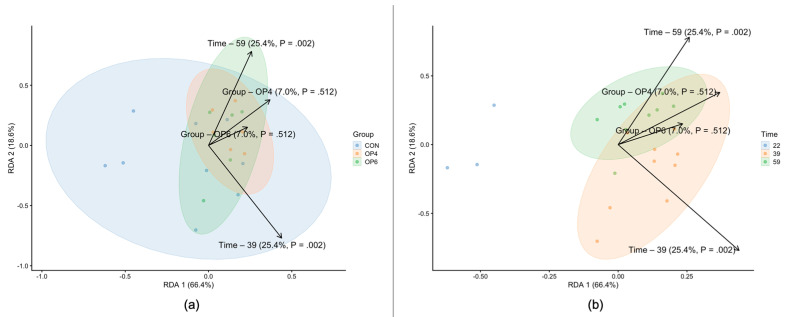
Results of the distance-based redundancy analysis (dbRDA) colored based on (**a**) treatment group (CON, OP4, and OP6) and by (**b**) time points (22, 39, and 59 weeks of age).

**Figure 4 microorganisms-12-01916-f004:**
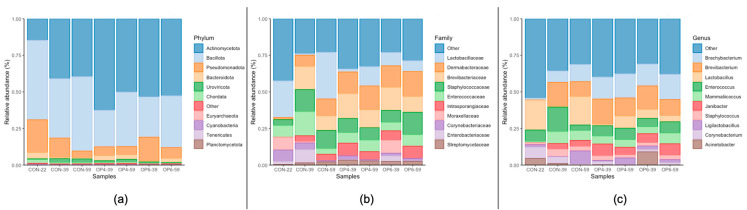
Relative abundance of the most frequent phyla (**a**), families (**b**), and genera (**c**) at different time points (22, 39, and 59 weeks of age) and treatment (CON, OP4, and OP6).

**Figure 5 microorganisms-12-01916-f005:**
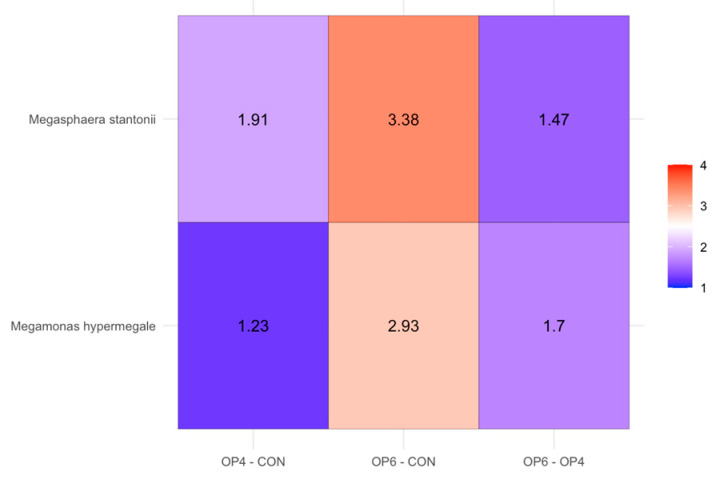
Species with increasing relative abundance per treatment. The values correspond to log fold changes between different treatment groups (CON, OP4, and OP6) at time point 59 weeks of age.

**Table 1 microorganisms-12-01916-t001:** Ingredient composition of the experimental diets (%).

Ingredients (%)	CON	OP4	OP6
Maize	56.6	59.0	57.0
Soya meal—47	25.5	24.8	24.8
Wheat bran	5.0	0	0
Olive pulp	0	4.0	6.0
Marble coarse	3.5	3.5	3.5
Marble powder	5.5	5.5	5.5
Vitamin and mineral premix ^1^	2.5	2.5	2.5
Soya oil	1.2	0.5	0.5
Salt	0.1	0.1	0.1
Sodium bicarbonate	0.1	0.1	0.1
Calculated analysis			
Lysine (%)	0.82	0.82	0.82
Methionine + Cystine (%)	0.65	0.65	0.65
Ca (%)	4.25	4.25	4.25
Av. P ^2^ (%)	0.34	0.34	0.34

^1^ Contains per Kg of product: vitamin A, 400,000 IU; vitamin D3, 120,000 IU; vitamin E, 1200 mg; vitamin K3, 160 mg; vitamin B1, 80 mg; vitamin B2, 240 mg; vitamin B6, 120 mg; vitamin B12, 0.8 mg; niacin, 1200 mg; D-calcium pantothenic acid, 480 mg; folic acid, 32 mg; biotin, 2 mg; choline chloride, 21,000 mg; vitamin C, 400 mg; zinc, 4000 mg; manganese, 4800 mg; iron, 2400 mg; copper, 400 mg; iodine, 80 mg; selenium, 8 mg; butilthydroxytoluen (BHT), 25 mg; phytase3, 3400 mg; chemical composition %: proteins, 5%; moisture, 5%; ash, 85%; calcium, 17%; phosphorus, 9.5%; lysine, 1%; methionine, 4%; sodium, 5%. ^2^ Av. P: available phosphorus.

**Table 2 microorganisms-12-01916-t002:** Nutritional analysis of the experimental diets. Data are presented as mean ± standard deviation (mean ± SD). They are derived from the analysis of three samples per batch in a total of three batches.

Parameter	CON	OP4	OP6
Energy (kJ/100 g)	1333.26 ± 34.93	1343.33 ± 24.86	1329.79 ± 38.40
Fat (%)	3.72 ± 0.27	3.51 ± 0.31	3.99 ± 0.10
SFA (%)	0.69 ± 0.16	0.60 ± 0.11	0.64 ± 0.01
MUFA (%)	1.25 ± 0.01	1.21 ± 0.13	1.30 ± 0.13
PUFA (%)	1.78 ± 0.16	1.70 ± 0.15	2.05 ± 0.19
Proteins (%)	17.50 ± 0.47	17.26 ± 0.52	17.29 ± 0.48
Carbohydrates (%)	51.74 ± 1.73	54.61 ±1.82	53.73 ± 1.24
Crude Fiber (%)	2.54 ± 1.16	3.03 ± 1.74	3.29 ± 1.49
Moisture (%)	9.54 ± 0.41	9.36 ± 1.05	9.49 ± 0.45
Ash (%)	14.96 ± 0.95	12.23 ± 1.34	12.21 ± 1.72
Total polyphenols (ppm)	95.40 ± 23.80	123.06 ± 37.66	137.24 ± 24.76
Cholesterol (ppm)	<10	<10	<10

**Table 3 microorganisms-12-01916-t003:** Nutritional analysis of the olive pulp used in the trial. Data are presented as mean ± standard deviation (mean ± SD). They are derived from the analysis of three samples per batch in a total of three batches.

Parameter	Olive Pulp
Energy (kJ/100 g)	1464.3 ± 22.01
Proteins (%)	8.5 ± 0.78
Carbohydrates (%)	40.2 ± 2.73
Crude Fiber (%)	29.3 ± 4.23
Moisture (%)	4.3 ± 0.23
Ash (%)	6.9 ± 1.97
Fat (%)	10.9 ± 0.56
Saturated Fatty Acids—SFA (%)	1.7 ± 0.17
Monounsaturated Fatty Acids—MUFA (%)	7.9 ± 0.45
Polyunsaturated Fatty Acids—PUFA (%)	1.3 ± 0.05
Total polyphenols (ppm)	573.70 ± 289.31
Cholesterol (ppm)	<10
Eleuropein (ppm)	20.7 ± 1.54
Hydroxytyrosol (ppm)	<3

**Table 4 microorganisms-12-01916-t004:** Fatty acid profile of olive pulp. Data are presented as mean ± standard deviation (mean ± SD). They are derived from the analysis of three samples per batch in a total of three batches.

Fatty Acids (g/100 g Fat)	
Lauric (dodecanoic) acid (C12:0)	0.03 ± 0.02
Myristic acid (C14:0)	0.05 ± 0.01
Palmitic acid (C16:0)	11.23 ± 0.56
Palmitoleic acid (C16:1)	0.57 ± 0.13
Margaric acid (C17:0)	0.12 ± 0.05
Cis-10-Heptadecenoic acid (C17:1)	0.19 ± 0.10
Stearic acid (C18: 0)	2.87 ± 0.05
Oleic acid (C18:1)	71.45 ± 0.44
α-Linoleic acid (C18:2)	10.47 ± 1.12
Linolenic acid(C18:3)	1.24 ± 0.12
Arachidic acid (C20:0)	0.56 ± 0.04
Arachidonic acid (C 20:4 ω6)	0.25 ± 0.14
Behenic acid (C22:0)	0.25 ± 0.05
Tricosanoic acid (C23:0)	0.15 ± 0.13
SFA	15.3 ± 0.78
MUFA	72.6 ± 0.39
PUFA	12.1 ± 1.13

**Table 5 microorganisms-12-01916-t005:** Body weight at week 59 (Final BW) and productive traits assessed during the trial. Data are presented as mean ± SE.

Items	CON	OP4	OP6
Final BW (Kg)	2.02 ± 0.02	2.05 ± 0.02	1.99 ± 0.02
HDEP (%)	94.31 ± 0.62	93.63 ± 0.56	94.30 ± 0.54
Egg weight (g)	64.25 ± 0.38	63.16 ± 0.40	63.80 ± 0.40
ADFI (g/h/d)	207.60 ± 5.17	210.61 ± 5.17	210.81 ± 5.01
Egg mass	60.83 ± 0.62	59.15 ± 0.54	60.18 ± 0.58
FCR	3.42 ± 0.08	3.57 ± 0.09	3.50 ± 0.07
Eggs with broken shell %	0.53 ± 0.08 ^a^	0.35 ± 0.05 ^b^	0.45 ± 0.07 ^ab^
Eggs with dirty eggshells %	4.80 ± 0.54	5.71 ± 0.56	6.36 ± 0.72

^a,b^ Means within a row with different superscripts differ significantly (*p* < 0.05).

**Table 6 microorganisms-12-01916-t006:** Egg quality traits were assessed in eggs collected from hens of all dietary treatments at 39 and 59 weeks of age. Group, age, and group × age effects are shown. Data are presented as marginal mean ± SE.

Parameter	Group	WK39	WK59	*Group Mean*	*P*		
					Group	Age	Group × Age
Egg weight	CON	63.25 ± 1.18	66.03 ± 1.18	*64.64* *± 0.83*	0.152	0.045	0.774
	OP4	61.42 ± 1.18	63.25 ± 1.18	*62.34* *± 0.83*			
	OP6	63.13 ± 1.18	64.22 ± 1.18	*63.68* *± 0.83*			
	*Age mean*	*62.60* ± *0.68 ^A^*	*64.50* *± 0.68 ^B^*				
Egg width	CON	44.24 ± 0.33	45.25 ± 0.33 ^a^	*44.74* *± 0.23 ^a^*	<0.001	0.721	<0.001
	OP4	44.21 ± 0.33 ^A^	42.53 ± 0.33 ^bB^	*43.37* *± 0.23 ^b^*			
	OP6	44.27 ± 0.33	44.65 ± 0.33 ^a^	*44.46* *± 0.23 ^a^*			
	*Age mean*	*44.24* ± 0.19	*44.14* ± 0.19				
Egg length	CON	56.69 ± 0.49	56.88 ± 0.49	*56.78 ± 0.34*	0.078	0.348	0.826
	OP4	55.39 ± 0.49	56.11 ± 0.49	*55.75 ± 0.34*			
	OP6	56.51 ± 0.49	56.73 ± 0.49	*56.62 ± 0.34*			
	*Age mean*	*56.20 ± 0.28*	*56.57 ± 0.28*				
Shape index (%)	CON	78.09 ± 0.76	79.65 ± 0.76 ^a^	*78.87 ± 0.54*	0.390	0.242	<0.001
	OP4	79.90 ± 0.76 ^A^	75.82 ± 0.76 ^bB^	*77.86 ± 0.54*			
	OP6	78.42 ± 0.76	78.76 ± 0.76 ^ab^	*78.59 ± 0.54*			
	*Age mean*	*78.80 ± 0.44*	*78.08 ± 0.44*				
SBS (N)	CON	52.68 ± 2.27	45.69 ± 2.27	*49.18 ± 1.61*	0.849	0.017	0.415
	OP4	51.54 ± 2.27	46.13 ± 2.27	*48.84 ± 1.61*			
	OP6	50.67 ± 2.27	49.53 ± 2.27	*50.10 ± 1.61*			
	*Age mean*	*51.63 ± 1.31 ^A^*	*47.12 ± 1.31 ^B^*				
Deformation (mm)	CON	0.95 ± 0.03	0.83 ± 0.03	*0.89 ± 0.02*	0.492	<0.001	0.400
	OP4	0.91 ± 0.03	0.79 ± 0.03	*0.85 ± 0.02*			
	OP6	0.88 ± 0.03	0.84 ± 0.03	*0.86 ± 0.02*			
	*Age mean*	*0.91 ± 0.02 ^A^*	*0.82 ± 0.02 ^B^*				
Albumen weight (gr)	CON	35.49 ± 0.89	36.47 ± 0.89	*35.98* *± 0.63*	0.212	0.873	0.620
	OP4	34.32 ± 0.89	34.46 ± 0.89	*34.39 ± 0.63*			
	OP6	35.58 ± 0.89	34.81 ± 0.89	*35.20 ± 0.63*			
	*Age mean*	*35.13* *± 0.51*	*35.25 ± 0.51*				
Albumen ratio (%)	CON	56.05 ± 0.66	55.23 ± 0.66	*55.64* *± 0.47*	0.700	0.007	0.631
	OP4	55.90 ± 0.66	54.33 ± 0.66	*55.12 ± 0.47*			
	OP6	56.24 ± 0.66	54.15 ± 0.66	*55.20 ± 0.47*			
	*Age mean*	*56.06 ± 0.38 ^A^*	*54.57 ± 0.38 ^B^*				
Yolk weight (gr)	CON	18.66 ± 0.41 ^a^	17.60 ± 0.41	*18.13 ± 0.29 ^a^*	<0.001	0.865	0.061
	OP4	15.58 ± 0.41 ^b^	16.32 ± 0.41	*15.95 ± 0.29 ^b^*			
	OP6	15.84 ± 0.41 ^b^	16.33 ± 0.41	*16.08 ± 0.29 ^b^*			
	*Age mean*	*16.69 ± 0.23*	*16.75 ± 0.23*				
Yolk ratio (%)	CON	29.58 ± 0.54 ^aA^	26.62 ± 0.54 ^B^	*28.10 ± 0.38 ^a^*	<0.001	0.123	0.002
	OP4	25.32 ± 0.54 ^b^	25.88 ± 0.54	*25.60 ± 0.38 ^b^*			
	OP6	25.14 ± 0.54 ^b^	25.48 ± 0.54	*25.31 ± 0.38 ^b^*			
	*Age mean*	*26.68 ± 0.31*	*25.99 ± 0.31*				
Shell weight (gr)	CON	7.65 ± 0.18	8.31 ± 0.18	*7.98 ± 0.12*	0.079	0.007	0.433
	OP4	7.57 ± 0.18	7.84 ± 0.18	*7.70 ± 0.12*			
	OP6	7.47 ± 0.18	7.73 ± 0.18	*7.60 ± 0.12*			
	*Age mean*	*7.56 ± 0.10 ^A^*	*7.96 ± 0.10 ^B^*				
Shell ratio (%)	CON	12.10 ± 0.22	12.59 ± 0.22	*12.34 ± 0.16*	0.100	0.141	0.634
	OP4	12.33 ± 0.22	12.42 ± 0.22	*12.38 ± 0.15*			
	OP6	11.85 ± 0.22	12.05 ± 0.22	*11.95 ± 0.16*			
	*Age mean*	*12.09 ± 0.13*	*12.36 ± 0.13*				
Albumen height (mm)	CON	8.31 ± 0.26	8.29 ± 0.26	*8.30 ± 0.19*	0.060	0.222	0.387
	OP4	7.42 ± 0.26	8.10 ± 0.26	*7.76 ± 0.19*			
	OP6	7.68 ± 0.26	7.82 ± 0.26	*7.75 ± 0.19*			
	*Age mean*	*7.80 ± 0.15*	*8.07 ± 0.15*				
Haugh unit	CON	90.05 ± 1.44	89.53 ± 1.44	*89.79 ± 1.02*	0.090	0.263	0.318
	OP4	85.36 ± 1.44	89.10 ± 1.44	*87.23 ± 1.02*			
	OP6	86.49 ± 1.44	87.23 ± 1.44	*86.86 ± 1.02*			
	*Age mean*	*87.30 ± 0.83*	*88.62 ± 0.83*				
Shell thickness (mm)	CON	0.69 ± 0.02 ^A^	0.44 ± 0.02 ^aB^	*0.56 ± 0.01 ^a^*	<0.001	<0.001	<0.001
	OP4	0.68 ± 0.02	0.66 ± 0.02 ^b^	*0.67 ± 0.01 ^b^*			
	OP6	0.68 ± 0.02	0.63 ± 0.02 ^b^	*0.66 ± 0.01 ^b^*			
	*Age mean*	*0.68 ± 0.01 ^A^*	*0.58 ± 0.01 ^B^*				
Yolk color	CON	6.80 ± 0.20 ^aA^	8.60 ± 0.20 ^B^	*7.70 ± 0.14 ^a^*	<0.001	<0.001	<0.001
	OP4	6.93 ± 0.20 ^aA^	8.93 ± 0.20 ^B^	*7.93 ± 0.14 ^a^*			
	OP6	9.20 ± 0.20 ^bA^	8.33 ± 0.20 ^B^	*8.77 ± 0.14 ^b^*			
	*Age mean*	*7.64 ± 0.12 ^A^*	*8.62 ± 0.12 ^B^*				
Yolk height (mm)	CON	18.66 ± 0.27	18.72 ± 0.27	*18.69 ± 0.20*	0.063	0.163	0.061
	OP4	18.06 ± 0.27	18.16 ± 0.27	*18.11 ± 0.20*			
	OP6	18.70 ± 0.27	17.63 ± 0.20	*18.16 ± 0.20*			
	*Age mean*	*18.48 ± 0.16*	*18.17 ± 0.16*				
Yolk width (mm)	CON	41.63 ± 0.47	42.84 ± 0.47	*42.23 ± 0.33*	0.224	<0.001	0.405
	OP4	40.33 ± 0.47 ^A^	42.53 ± 0.47 ^B^	*41.43 ± 0.33*			
	OP6	41.43 ± 0.47	42.45 ± 0.47	*41.94 ± 0.33*			
	*Age mean*	*41.13 ± 0.27 ^A^*	*42.61 ± 0.27 ^B^*				
Yolk index	CON	0.45 ± 0.01	0.44 ± 0.01	*0.44 ± 0.01*	0.538	<0.001	0.291
	OP4	0.45 ± 0.01	0.43 ± 0.01	*0.44 ± 0.01*			
	OP6	0.45 ± 0.01 ^A^	0.42 ± 0.01 ^B^	*0.43 ± 0.01*			
	*Age mean*	*0.45 ± 0.01 ^A^*	*0.43 ± 0.01 ^B^*				
Albumen pH	CON	8.74 ± 0.07 ^A^	8.30 ± 0.07 ^Βa^	*8.52 ± 0.05 ^a^*	<0.001	<0.001	<0.001
	OP4	8.90 ± 0.07 ^A^	8.39 ± 0.07 ^Βa^	*8.64 ± 0.05 ^a^*			
	OP6	8.80 ± 0.07	8.84 ± 0.07 ^b^	*8.82 ± 0.05 ^b^*			
	*Age mean*	*8.81 ± 0.04 ^A^*	*8.51 ± 0.04 ^B^*				
Yolk pH	CON	6.04 ± 0.09	5.82 ± 0.09	*5.93 ± 0.06*	0.248	<0.001	0.445
	OP4	6.28 ± 0.09 ^A^	5.87 ± 0.09 ^B^	*6.08 ± 0.06*			
	OP6	6.11 ± 0.09	5.90 ± 0.09	*6.01 ± 0.06*			
	*Age mean*	*6.15 ± 0.05 ^A^*	*5.86 ± 0.06 ^B^*				

^a,b^ Means within columns differ significantly (*p* < 0.05). ^A,B^ Means within rows differ significantly (*p* < 0.05).

**Table 7 microorganisms-12-01916-t007:** Biochemical parameters assessed in blood samples collected from hens of all groups at 39 and 59 weeks of age. Group, age, and group × age effects are shown. Data are presented as marginal mean ± SE.

Parameter	Group	WK39	WK59	*Group Mean*	*P*		
					Group	Age	Group × Age
Cholesterol (mg/dL)	CON	124.07 ± 8.56	117.20 ± 8.56	*120.63 ± 6.05*	0.895	0.065	0.205
	OP4	133.81 ± 8.29	103.47 ± 8.56	*118.64 ± 5.96*			
	OP6	123.53 ± 8.56	121.73 ± 8.56	*122.63 ± 6.05*			
	*Age mean*	*127.14 ± 4.89*	*114.13 ± 4.94*				
Triglycerides (mg/dL)	CON	1778.93 ± 193.39	1762.33 ± 193.39	*1770.63 ± 136.75*	0.833	0.327	0.533
	OP4	1867.19 ± 187.25	1462.60 ± 193.39	*1664.89 ± 134.60*			
	OP6	1781.87 ± 193.39	1735.86 ± 200.18	*1758.86 ± 139.17*			
	*Age mean*	*1809.33 ± 110.49*	*1653.60 ± 112.98*				
AST (IU/L)	CON	131.00 ± 3.81 ^A^	114.40 ± 3.81 ^B^	*122.70 ± 2.69*	0.095	<0.001	0.086
	OP4	133.44 ± 3.69	125.93 ± 3.81	*129.69 ± 2.65*			
	OP6	142.40 ± 3.81 ^A^	117.93 ± 3.81 ^B^	*130.17 ± 2.69*			
	*Age mean*	*135.61 ± 2.18 ^A^*	*119.42 ± 2.20 ^B^*				
G-GT (IU/L)	CON	25.53 ± 2.66 ^aA^	36.93 ± 2.66 ^B^	*31.23 ± 1.88 ^a^*	<0.001	0.781	0.002
	OP4	43.06 ± 2.57 ^b^	35.93 ± 2.66	*39.50 ± 1.85 ^b^*			
	OP6	42.00 ± 2.66 ^b^	39.53 ± 2.66	*40.77 ± 1.88 ^b^*			
	*Age mean*	*36.87 ± 1.52*	*37.47 ± 1.53*				
Uric acid (mg/dL)	CON	7.88 ± 0.35	7.00 ± 0.36	*7.44 ± 0.26*	0.210	<0.001	0.440
	OP4	7.66 ± 0.34 ^A^	6.04 ± 0.35 ^B^	*6.85 ± 0.24*			
	OP6	7.81 ± 0.35 ^A^	6.08 ± 0.35 ^B^	*6.94 ± 0.25*			
	*Age mean*	*7.78 ± 0.20 ^A^*	*6.37 ± 0.21 ^B^*				
BUN (mg/dL)	CON	11.00 ± 0.83	11.47 ± 0.83	*11.23 ± 0.59*	0.749	0.734	0.564
	OP4	11.63 ± 0.81	10.40 ± 0.83	*11.01 ± 0.58*			
	OP6	11.60 ± 0.83	11.67 ± 0.83	*11.63 ± 0.59*			
	*Age mean*	*11.41 ± 0.48*	*11.18 ± 0.48*				
GLDH (U/L)	CON	4.07 ± 0.34	2.67 ± 0.34	*3.37 ± 0.24 ^a^*	0.014	0.001	0.519
	OP4	4.75 ± 0.34	3.93 ± 0.35	*4.34 ± 0.24 ^b^*			
	OP6	3.87 ± 0.34	3.23 ± 0.37	*3.55 ± 0.25 ^ab^*			
	*Age mean*	*4.23 ± 0.20 ^A^*	*3.28 ± 0.21 ^B^*				

^a,b^ Means within columns differ significantly (*p* < 0.05). ^A,B^ Means within rows differ significantly (*p* < 0.05).

## Data Availability

The data presented in this study are available from the corresponding author on request. The data are not publicly available due to privacy reasons.

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
