# Peer review of "The Influence of Dietary Supplementation with Dried Olive Pulp on Gut Microbiota, Production Performance, Egg Quality Traits, and Health of Laying Hens"

_microorganisms, 2024, doi:10.3390/microorganisms12091916_

Round 1

Reviewer 1 Report

Comments and Suggestions for Authors

Reviewer 2 Report

Comments and Suggestions for Authors

Alteration of gut microbiota in laying hens fed diets supplemented with dried olive pulp and its effect on production performance, egg quality and birds’ health.

Very good research and well executed. 

The manuscript is too long, try to shorten the introduction and discussion parts.

Add statistical design to the abstract.

Add P value to the abstract.

L119: what was the purpose of the adaptation period?

Table 2 and 3: provide briefly how did you measure these parameters in the material and methods section.

Provide the source of OP, how it was prepared? does it contain stones?

You need to explain the 2-way interactions with more details. I prefer providing small graphs for all significant interactions

Reviewer 3 Report

Comments and Suggestions for Authors

Dear authors 

Thanks for your effort and presentation.

Some points could be considered during your revision.

1. The title could be modified according to your aim of work to the following;

The influence of dietary dried olive pulb supplementation on the gut macrobiota, production performance,  egg traits, and health of layer chickens.

2. The keywords should be a little bit different from those of the title. 

3. Most of the mentioned parameters didn't contain references.

4. Please explain what are the added value of your work, especially you mentioned a lot of similar previous similar work.

5. The conclusion is so long and contains many explanations that could be represented in the discussion section. So, please re-write it in a concise form with some recommendations. 

Best wishes 
